# Characterization of Plasma-Derived Small Extracellular Vesicles Indicates Ongoing Endothelial and Platelet Activation in Patients with Thrombotic Antiphospholipid Syndrome

**DOI:** 10.3390/cells9051211

**Published:** 2020-05-13

**Authors:** Ula Štok, Elizabeta Blokar, Metka Lenassi, Marija Holcar, Mojca Frank-Bertoncelj, Andreja Erman, Nataša Resnik, Snežna Sodin-Šemrl, Saša Čučnik, Katja Perdan Pirkmajer, Aleš Ambrožič, Polona Žigon

**Affiliations:** 1Department of Rheumatology, University Medical Centre Ljubljana, SI-1000 Ljubljana, Slovenia; ulastok@gmail.com (U.Š.); elizabeta.blokar@gmail.com (E.B.); ssodin1@yahoo.com (S.S.-Š.); sasa.cucnik@kclj.si (S.Č.); katja.perdan@mf.uni-lj.si (K.P.P.); ales.ambrozic@mf.uni-lj.si (A.A.); 2Faculty of Pharmacy, University of Ljubljana, SI-1000 Ljubljana, Slovenia; 3Division for Internal Medicine, Faculty of Medicine, University of Ljubljana, SI-1000 Ljubljana, Slovenia; 4Institute of Biochemistry, Faculty of Medicine, University of Ljubljana, SI-1000 Ljubljana, Slovenia; metka.lenassi@mf.uni-lj.si (M.L.); marija.holcar@mf.uni-lj.si (M.H.); 5Centre of Experimental Rheumatology, Department of Rheumatology, University Hospital Zurich, 8952 Schlieren, Switzerland; mojca.frankbertoncelj@usz.ch; 6Institute of Cell Biology, Faculty of Medicine, University of Ljubljana, SI-1000 Ljubljana, Slovenia; andreja.erman@mf.uni-lj.si (A.E.); natasa.resnik@mf.uni-lj.si (N.R.); 7Faculty of Mathematics, Natural Sciences and Information Technologies, University of Primorska, SI-6000 Koper, Slovenia

**Keywords:** antiphospholipid syndrome, thrombosis, small extracellular vesicles, surface protein markers, endothelial activation, platelet activation, adhesion molecules

## Abstract

Antiphospholipid syndrome (APS) is a systemic autoimmune disease, characterized by thrombosis, obstetric complications and the presence of antiphospholipid antibodies (aPL), which drive endothelial injury and thrombophilia. Extracellular vesicles (EVs) have been implicated in endothelial and thrombotic pathologies. Here, we characterized the quantity, cellular origin and the surface expression of biologically active molecules in small EVs (sEVs) isolated from the plasma of thrombotic APS patients (*n* = 14), aPL-negative patients with idiopathic thrombosis (aPL-neg IT, *n* = 5) and healthy blood donors (HBD, *n* = 7). Nanoparticle tracking analysis showed similar sEV sizes (110–170 nm) between the groups, with an increased quantity of sEVs in patients with APS and aPL-neg IT compared to HBD. MACSPlex analysis of 37 different sEV surface markers showed endothelial (CD31), platelet (CD41b and CD42a), leukocyte (CD45), CD8 lymphocyte and APC (HLA-ABC) cell-derived sEVs. Except for CD8, these molecules were comparably expressed in all study groups. sEVs from APS patients were specifically enriched in surface expression of CD62P, suggesting endothelial and platelet activation in APS. Additionally, APS patients exhibited increased CD133/1 expression compared to aPL-neg IT, suggesting endothelial damage in APS patients. These findings demonstrate enhanced shedding, and distinct biological properties of sEVs in thrombotic APS.

## 1. Introduction

Antiphospholipid syndrome (APS) is a systemic autoimmune disease characterized by thromboses and/or obstetric complications, as confirmed with the presence of antiphospholipid antibodies (aPL) on at least two occasions, 12 weeks apart [1]. Associated clinical manifestations may include livedo reticularis, cutaneous ulcerations, thrombocytopenia, haemolytic anaemia, valvular heart disease and nephropathy, among others [2]. Laboratory criteria for APS include testing for lupus anticoagulant (LA), moderately high titre anti-cardiolipin (aCL) or anti-β2 glycoprotein (anti-β2GPI) antibodies. The presence of non-criteria antibodies, such as anti-phosphatidylserine/ prothrombin antibodies (aPS/PT), may increase the risk of clinical manifestations of APS [3]. The degree of risk associated with aPL depends not only on the characteristics of their profile, but also on the presence of additional thrombotic risk factors, such as hyperlipidaemia, arterial hypertension, diabetes, smoking, obesity, low grade inflammation and use of contraceptives [1]. Despite some improvements in the diagnosis and prognosis of APS and prevention of thrombosis reoccurrence, robust laboratory biomarkers are still lacking.

Communication and networking between cells are important mechanisms of interaction between healthy and pathologically modified cells. Despite the well-characterized in vitro and in vivo models of APS pathology in terms of aPL, the study area of extracellular vesicles (EVs) is largely unexplored, and could provide insight into the APS mechanism and possibly serve as biomarkers to identify patients at greater risk [4]. It is known that aPL, particularly anti-β2GPI, cause activation of vascular cells, such as endothelial cells, platelets and monocytes, which may result in the release of EVs. EVs can be divided into subsets based on their size, namely “small EVs” (sEVs; size < 100 nm or < 200 nm) and “medium/large EVs” (size > 200 nm), with their biogenesis reflected in their molecular composition (e.g., using their surface proteins) [5]. EVs are secreted by different cell types and detected in different bodily fluids, including peripheral blood. They contain proteins, nucleic acids (e.g., miRNA, mRNA, lncRNA, circRNA) and lipids, generally reflecting the status of the parental cell. sEVs have increasingly attracted attention for their role in physiology and pathology, as well as their possible use as diagnostic and therapeutic tools. Especially attractive features of sEVs are their different biophysical/biochemical properties, as compared to soluble markers, making their lifespan longer and more resistant to protease degradation [6]. The few studies that have examined EVs in APS have focused primarily on medium/large EVs [7,8,9,10,11,12,13,14]. To date, there are no studies available on sEVs in patients with APS. The purpose of this study was to characterize the sEVs from APS patients in terms of their concentration, size and selected surface proteins, as well as to identify the potential differences from healthy blood donors (HBD) and aPL-negative patients with idiopathic thrombosis (aPL-neg IT). 

## 2. Materials and Methods

### 2.1. Patients and Controls

Patients were recruited from the Department of Rheumatology, University Medical Centre Ljubljana. All 19 patients included in this study experienced one or more thrombotic events (arterial and/or venous thrombosis, with/without obstetric complication) with no acute thrombosis at the time of blood collection. None of the patients experienced only obstetric complications. Fourteen patients fulfilled the updated international classification criteria for APS [1], while 5 patients were negative for all the classes of thrombosis-associated aPL (aCL, anti-β2GPI, aPS/PT and LA). aPL-neg IT were diagnosed as suffering from idiopathic thrombosis and were enrolled in the study as a control thrombotic group, repeatedly negative for the presence of aPL (Table 1). The study also included 7 age-matched HBD. 

At the time of the visit, a participant’s medical history was recorded for venous, arterial or micro thrombosis, as well as for history of obstetric complications and diabetes. Treatment status was recorded (e.g., anticoagulation, anti-aggregation, antimalaric therapy), including oral contraception (current/at thrombotic event). A number of parameters that could confound the EV determination and characteristics were recorded at the time of obtaining the blood samples, as recommended by the International Society for Extracellular Vesicles (ISEV) [5]. These variables included age, gender, body mass index (BMI), smoking status, fasting status, systolic pressure and diastolic pressure.

This study was approved by the National Medical Ethics Committee, Ljubljana, Slovenia (0120-7/2019/5). All participants provided informed consent according to the Declaration of Helsinki. 

### 2.2. Blood Collection 

Serum and citrated plasma were obtained from the whole blood of patients and HBD (Figure 1). Citrated plasma was divided and used for the analysis of LA and isolation of sEVs. Serum was used for measurements of aPL and other biochemical factors described below. All samples were processed within one hour of blood drawing. Serum tubes were kept at room temperature for 30 min before centrifugation at 1800 × g for 10 min at RT (1624, Universal 320 R, Hettich, Tuttlingen, Germany). 

### 2.3. Biochemical Analysis

We analysed the complete blood counts with an Advia Hematology 120 (Simens Healthineers, Erlangen, Germany); the erythrocyte sedimentation rate (ESR) by the WesternGreen method for 1 h; serum amyloid A (SAA) by nephelometry (Atellica NEPH 630, Simens Healthineers, Erlangen, Germany); C-reactive protein (CRP) by immunoturbidimetry; glucose by the glucose hexokinase method; cholesterol by cholesterol enzymatic colorimetric CHOD-PAP; high density lipoproteins (HDL) by HDL elimination/catalase; triglycerides by triglyceride enzymatic colorimetric GPO-PAP (all using Advia 1800 Chemistry System, Simens Healthlineers, Erlangen, Germany); and low density lipoproteins (LDL) by calculation from cholesterol, HDL and triglycerides. All tests were performed as recommended by the manufacturer. These parameters were measured as they could importantly confound EV determination or characteristics in accordance with the ISEV recommendations [5]. 

### 2.4. aPL Determination

Patient sera were measured for an aPL profile, including LA, aCL, anti-β2GPI, aPS/PT of IgG, IgM and IgA isotypes, using our in-house aCL [15], anti-β2GPI [16] and aPS/PT [17] ELISAs, as previously described. For determining LA, platelet-poor plasma was obtained by centrifugation at 2.000 × g for 15 min at 15 °C (1624, Universal 320 R, Hettich, Tuttlingen, Germany). After filtration, aliquots were stored at −80 °C until use. Clotting tests were performed using a coagulation analyser CS-2500 (Sysmex, Kobe, Japan) and STart (Diagnostica Stago, Asnières sur Seine Cedex, France), according to the previous guidelines of the International Society on Thrombosis and Haemostasis ISTH. We performed the Dilute Russell’s Viper Venom Test (dRVVT) test with the LA1 screening reagent and LA2 confirmatory reagent (Siemens Healthlineers, Erlangen, Germany) following the manufacturer’s instructions. Staclot LA (Diagnostica Stago, Asnières sur Seine Cedex, France) was used for LA detection/confirmation.

### 2.5. Preparation of Plasma for EV Isolation

For isolation of EVs from plasma, we followed the ISEV recommendations [5]. Plasma was separated from blood cells within one hour after blood drawing by centrifugation at 820 × g for 10 min at room temperature (1624, Universal 320 R, Hettich, Tuttlingen, Germany). In the second centrifugation step, large debris/larger particles were removed by centrifuging plasma at 2500 × g for 10 min to obtain platelet-poor plasma. One mL of the platelet-poor plasma intended for later isolation of sEVs by Sucrose Cushion Ultracentrifugation was stored at −80 °C. The remaining plasma was further centrifuged at 10,000 × g for 45 min at room temperature to pellet small debris/larger EVs (IL 085, 5430R, Eppendorf, Hamburg, Germany), and stored at 4 °C for isolation of the sEVs using CD63 Immuno-Magnetic beads. 

### 2.6. Sucrose Cushion Ultracentrifugation

One mL of platelet-poor plasma was thawed on ice and centrifuged at 10,000 × g for 20 min at 4 °C. Next, the supernatant was diluted to 9 mL with dPBS and carefully pipetted over 2 mL of the 20% sucrose cushion (Merck Millipore, Burlington, MA, USA) in dPBS in 13 mL polypropylene ultracentrifuge tubes (Beckman Coulter, Brea, CA, USA). After ultracentrifugation at 100,000 × g for 2 h 15 min at 4 °C, the supernatant was removed and the pellet was resuspended in 40 μL of dPBS, transferred to a fresh protein low binding tube (Eppendorf, Hamburg, Germany) and stored at −20 °C. The sucrose cushion ultracentrifugation is an adapted protocol of the original ultracentrifugation method (internal communication with Lenassi M. and Holcar M.). 

### 2.7. Quantification of the Concentration and Size of the sEVs 

The concentration and size of the sEVs were determined by nanoparticle tracking analysis (NTA) using the NanoSight NS300 instrument (488 nm laser) connected to a sample assistant (both Malvern Panalytical, Malvern, UK) for automated sample processing. Samples were diluted 200× in dPBS to obtain a particle concentration between 1 × 10^7^/mL and 1 × 10^9^/mL. For each sample, five 60 s videos were recorded at camera level 14, visually inspected and excluded from the analysis if any major distortion was detected. Raw data from at least three videos per sample were analysed by the NanoSight NTA 3.3 program at the following settings: a detection threshold of 5, water viscosity approximation for dPBS, a temperature of 25 °C, automatic settings for minimum expected particle size and blur as well as a minimum track length of 10.

### 2.8. CD63 Immunomagnetic Isolation of sEVs

For CD63-based Immuno-Magnetic Isolation, fresh platelet-poor plasma was used, centrifuged at 10,000 × g for 45 min at RT (IL 085, 5430R, Eppendorf, Hamburg, Germany), and the supernatants were stored at 4 °C and used for CD63-based isolation in the next three days. One mL of plasma was diluted 1:1 in 1x sterile dPBS (Lonza, Basel, Switzerland). Isolation of sEVs was performed using a CD63 Exosome Isolation Kit (Miltenyi Biotech, Bergisch Gladbach, Germany) following the manufacturer’s protocol. Briefly, 50 µL of CD63 Exosome Isolation Microbeads were added to the diluted sample, incubated for 1 h at room temperature and then loaded onto a µ column placed on the magnetic stand for positive selection of the sEVs complexed to CD63 magnetic beads. For elution of the complexed sEVs the µ column was removed from the magnetic field and the complexes were eluted with 100 µL of Elution Buffer by firmly pushing the plunger into the column. 

### 2.9. Multiplex Bead-Based Flow Cytometry Analysis of the sEVs

We used the MACSPlex Exosome Kit, which allows detection of 37 membrane surface epitopes (CD1c, CD2, CD3, CD4, CD8, CD9, CD11c, CD14, CD19, CD20, CD24, CD25, CD29, CD31, CD40, CD41b, CD42a, CD44, CD45, CD49e, CD56, CD62P, CD63, CD69, CD81, CD86, CD105, CD133/1, CD142, CD146, CD209, CD326, HLA-ABC, HLA-DRDPDQ, MCSP, ROR1 and SSEA-4) and included two isotype controls (mIgG1 and REA), corresponding to the antibodies used. MACSPlex Capture beads are color-coded and will arrange in separate positions when looking at the PE-FITC channels. These beads are coated with antibodies of different epitope specificity. Particles binding to the beads are detected by measuring the APC signal of the detection antibodies against tetraspanins (aCD9 and aCD81), which are specific for sEVs. Briefly, sEVs–CD63 magnetic bead complexes were incubated with 15 µL of MACSPlex Capture Beads and incubated overnight protected from light on an orbital shaker at 450 rpm at RT. Next, samples were incubated with 5 µL of APC-conjugated detection antibodies directed against CD9 and CD81 for 1 h at RT protected from light on an orbital shaker at 450 rpm. After incubation, samples were washed and the APC signal intensity in each of the 39 specific bead populations was measured on a MACSQuant^®^ Analyzer 10 and analysis performed using the MACSQuant^®^ Analyzer Express Mode and MACSQuantify™ Software version 2.11 (Miltenyi Biotech, Bergisch Gladbach, Germany). Median fluorescence intensities (MFI) for all the capture beads were corrected for background signal by subtracting the respective MFI values from the non-EV containing buffer sample included in every analysis. The measured MFI inside each gate of a separate bead population was normalized to combined mean tetraspanin CD9/CD81 MFI values in order to determine the relative levels of a surface marker. 

### 2.10. Transmission Electron Microscopy

The sEVs, pelleted with sucrose cushion ultracentrifugation, were resuspended in dPBS and 5 µL of the suspension were deposited on copper Formvar-coated grids. Adsorbed sEVs were fixed with 1% glutaraldehyde, stained with 1% aqueous uranyl acetate, and air-dried. Samples were examined with a Philips CM100 transmission electron microscope operated at 80 kV. Images were captured with an AMT camera (Advanced Microscopy Techniques Corp., Woburn, MA, USA).

### 2.11. Statistical Analysis

Statistical analyses were performed using IBM SPPS Statistics, version 20. A Kruskal–Wallis test with Dunn’s multiple comparisons adjustment was used to compare the markers between the study groups. The χ^2^ test was used for categorical variables.

## 3. Results

### 3.1. Subject Characteristics That Could Confound the sEV Analysis Were Mostly Comparable between Study Groups 

Characteristics of patients and HBD enrolled in this study are presented in Table 1. A similar percentage of thrombosis was observed between APS patients and aPL-neg IT patients. While every APS patient tested positive for at least one class of aPL, HBD and aPL-neg IT tested negative for all aPL types. The biochemical variables that were measured (e.g., glucose, cholesterol, HDL, LDL, triglyceride, SAA, CRP, ESR, platelets), however, showed no significant differences between the groups (data not shown). Other known thrombotic risk factors, including smoking, BMI, hyperlipidaemia and diabetes, did not differ among the study groups. Patients with APS had significantly higher systolic (*p* = 0.005) and diastolic blood pressure (*p* = 0.029) but did not differ in other demographic, clinical and laboratory parameters, such as use of hormonal contraceptives, nor in fasting status compared to both control groups. A significant difference between APS and aPL-neg IT groups was observed in anticoagulant therapy (*p* = 0.005), while the use of anti-aggregation and antimalarial drugs was similar between these two groups. These data show that most of the tested confounding parameters that could affect the determination and characteristics of sEVs in plasma isolates did not differ among the study groups, except for the use of anticoagulant drugs and blood pressure, which were present and increased in APS patients and could potentially influence the release of sEVs. 

### 3.2. The Quantity of sEVs Is Increased in Patients with APS-Associated Thrombosis and aPL-Negative Idiopathic Thrombosis

We used NTA to determine the concentration and size of the sEVs isolated from the plasma of the study subjects by sucrose cushion ultracentrifugation (Figure 1). Significantly elevated numbers of sEVs (mean 4.95 ×10^9^ per 1 mL of plasma, *p* = 0.0206) were isolated from the plasma of patients with APS as well as aPL-neg IT (5.88 ×10^9^ per mL, *p* = 0.0067) compared to 2.64 ×10^9^ isolated sEVs per mL of plasma from HBD (Figure 2A). These data suggested that APS patients and patients with a history of idiopathic thrombosis show increased cell membrane vesiculation, even in the absence of an acute thrombotic event. The isolated sEVs showed similar sizes between the study groups (Figure 2B,C).

### 3.3. Transmission Electron Microscopy Confirms the Presence of sEVs in Plasma Isolates

The presence and purity of the sEVs in the plasma isolates from HBD, APS patients and aPL-neg IT patients were confirmed using transmission electron microscopy. We observed the characteristic concave shape of the EVs with sizes from 100 to 200 nm that did not differ between study groups. The electron microscopy analysis of the sEVs confirmed the size measurements of the sEVs with NTA, indicating the robustness of our analyses (Figure 3).

### 3.4. MACSPlex Technology Enables Detection of Multiple Surface Protein Markers on sEVs Isolated from Human Plasma 

Using the MACSPlex Assay coupled with flow cytometry, we were able to compare the expression of surface protein markers on CD63-positive sEVs isolated from plasma of patients with APS, aPL-neg IT and HBD. The MACSPlex platform comprises 37 surface protein capture beads (Appendix A) and two isotype control beads to determine the unspecific binding of the sEVs. Binding of the sEVs to a specific capture bead was detected with APC-conjugated tetraspanins (e.g., CD9 and CD81). In our study, CD63 could not be used for the detection, since it was occupied by magnetic beads from the sEV isolation procedure. MACSPlex enabled the detection of cell identity markers, as well as distinct biologically active surface molecules, covering a broad range of biological functions that are relevant for pathogenesis of APS (e.g., cell-to-cell adhesion, adhesion to the extracellular matrix, endothelial and platelet activation, and coagulation). The mean fluorescence intensity (MFI) of 17 out of 37 measured markers, including CD1c, CD2, CD3, CD4, CD11c, CD14, CD19, CD20, CD25, CD56, CD86, CD105, CD142, CD209, MCSP, SSEA-4 and ROR1, was smaller than the MFI of the isotype controls, suggesting their low expression (below the detection limit of the assay) or even absence in all three study groups. The remaining 17 markers (CD8, CD24, CD29, CD31, CD40, CD41b, CD42a, CD44, CD45, CD49e, CD62P, CD69, CD133/1, CD146, CD326, HLA-ABC and HLA-DRDPDQ) exhibited an MFI higher than the corresponding isotype control, demonstrating their presence on the sEVs (Figure 4 and Appendix A). The signal intensities of the detected markers were normalized to the mean signal intensity of the CD9 and CD81 tetraspanins. We clustered the detected 17 sEVs surface protein markers into markers of the cell origin (CD8, CD24, CD41b, CD42a, CD45, CD31 and HLA-ABC) and markers informing of the cell activation or functional status of the sEVs (CD40, CD62P, CD69, CD133/1 and HLA-DRDPDQ, and CD29, CD44, CD49e, CD146 and CD326, respectively).

### 3.5. Platelet-, Lymphocyte-, Leukocyte- and Endothelial-Derived sEVs are Present in Plasma of Patients with APS, aPL-Negative Idiopathic Thrombosis and Healthy Blood Donors

In the plasma samples from the study subjects, we detected sEVs originating from various hematopoietic cells. In addition to erythrocytes, platelets are the most abundant component of the peripheral blood, which readily shed EVs also ex vivo after blood collection. sEVs isolated from the plasma of the study subjects exhibited the expression of platelet markers, specifically CD41b and CD42a (Figure 4, upper panel). There was no difference in the surface expression of these markers between the three study groups, indicating comparable in vivo and ex vivo platelet activation and likely reflecting reduced pre-analytical variability in blood sampling and processing.

Among the lymphocyte markers, we detected the expression of CD8 on sEVs. Notably, CD8 expression was enriched on the surface of sEVs from plasma of APS patients as compared to aPL-neg IT (*p* = 0.015), potentially reflecting an activation of the immune system in autoimmune APS patients (Figure 4, upper panel).

Endothelial-derived and leukocyte-derived sEVs, as detected by surface expression of the endothelial cell marker CD31 and pan-leukocyte marker CD45, were present in our samples with nonsignificant changes between the study groups (Appendix A). Among other cell type-specific markers, CD24 was detected on the surface of sEVs with nonsignificant changes in expression between the study groups (Appendix A). CD24 is expressed on most B-cells, but it is not a B-cell specific marker, since it is also expressed on neutrophils and differentiating neuroblasts. Other B cell-markers, including CD19 and CD20, were not detected on the surface of isolated plasma sEVs and the same was true for the markers of monocytes and dendritic cells, including CD14, CD80, CD86 and CD209. However, HLA-ABC-positive sEVs could be detected in our samples, suggesting their origin from antigen-presenting cells (Appendix A).

### 3.6. sEVs from Plasma of Patients with APS Are Enriched for the Surface Expression of CD62P and CD133/1, Indicating Endothelial Activation/Damage and Platelet Activation in APS 

The analysis of the surface CD62P expression on sEVs showed enriched presence of this marker on the surface of sEVs from patients with APS as compared to HBD (*p* = 0.019) (Figure 4, lower panel). CD62P is a platelet and endothelial cell activation marker, indicating the ongoing platelet and endothelial activation in APS, even in the absence of acute thrombotic events. Furthermore, sEVs from patients with APS expressed increased surface levels of CD113/1 compared to aPL-neg IT (*p* = 0.016) (Figure 4, lower panel). CD113/1 is expressed on different cell types, including endothelial progenitor cells. APS patients are known to exhibit an impaired endothelium function due to endothelial activation and injury. sEVs might reflect an ongoing endothelial repair mechanism in patients with APS.

Amidst other expressed immune-cell-related surface proteins on isolated sEVs from plasma of the three study groups, we detected the expression of CD40, CD69 and HLA-DRDPDQ with insignificant changes between the groups (Appendix A). CD40, CD69 and HLA-DRDPDQ are markers of B-cell activation, thus suggesting the activated state of the B-cells in the study groups. 

### 3.7. EVs Isolated from Human Plasma Express Several Adhesion Molecules Involved in Cell-to-Cell and Cell-to-Matrix Interactions

We detected populations of sEVs expressing the CD44 cell-to-cell adhesion molecule. Endothelial cells, immune cells and platelets express higher levels of adhesion molecules when activated, increasing their stickiness, which is an important risk factor for thrombosis. Notably, the expression of CD44 was decreased (*p* = 0.031) in aPL-neg IT compared to patients with APS (Figure 4, lower panel). CD44 is expressed on a variety of cells and serves as an adhesion molecule for extracellular matrix components (hyaluronic acid, collagens) and matrix metalloproteinases. CD44 has multiple functions, including lymphocyte activation, recirculation, homing, haematopoiesis and tumour metastasis. Its decreased expression on the surface of aPL-neg IT might reflect the lower adhesion potential of these sEVs. 

sEVs expressing cell-to-cell adhesion molecules (CD146 and CD326) and cell-to-extracellular matrix adhesion molecules (CD29 and CD49e) were observed with insignificant changes between the study groups (Appendix A). 

## 4. Discussion

The last decade has seen a sharp increase in the number of studies investigating the physiological and pathological functions of EVs. Surface protein profiles of larger EVs have been studied in APS patients [4] but to date, there has been no information on sEVs with sizes <200 nm, and their role in the pathogenesis of APS. 

Our study enrolled a relatively small number of subjects; they, however, were clinically well characterized and confounding variables in the EV characterization were comprehensively determined. To minimize potential differences between the study groups that could affect sEVs’ concentration and properties, we followed carefully the protocols recommended by the ISEV [5]. Most confounding variables did not significantly differ between the study groups, except for systolic and diastolic blood pressure levels and anticoagulant treatment. The latter might contribute to altered release of sEVs between study groups. Arterial hypertension has been described as one of the clinical characteristics in APS patients [1] and has been defined as a risk factor for recurrent thrombotic events in APS [18].

The number of sEVs in healthy human plasma varies between 10^11^ and 10^12^ per mL [19]. This variation in sEV quantities can reflect the significant influence of different pre-analytical variables during blood collection and EV isolation on EV measurements, as well as variability in methodologies used for isolation and detection of sEVs. Plasma sEV isolates could also be contaminated with lipoproteins and protein aggregates. Most quantification methods, including NTA, are unable to distinguish these particles from sEVs. Thus, isolation methods enabling separation of sEVs from these particles are preferred. Among many different techniques that have been developed for isolation of sEVs from plasma, ultracentrifugation remains the golden standard, according to the ISEV 2018 guidelines [5]. In our study, we used an adapted sucrose cushion protocol for ultracentrifugation, resulting in highly pure sEVs isolates largely devoid of lipoproteins and protein aggregates. This strongly facilitated the accuracy of quantification of sEVs by NTA. Our stringent isolation/purification protocol likely explains why the concentrations of sEVs in our study are lower than described in other studies [19]. 

In our study, we included both patients with APS and aPL-neg IT. These two groups would likely differ in sEVs since the mechanisms of thrombosis and the immune system activation differ between these conditions. By including the aPL-neg IT patient group, we were able to investigate whether aPL have a role in sEVs release or influence their surface protein profile. We demonstrated that patients with APS and aPL-neg IT have increased quantities of sEVs in plasma isolates compared to HBD, suggesting an enhanced shedding of cellular membranes in these subjects, also in the absence of acute thrombotic events. This might contribute to increased susceptibility of these patients to developing thrombosis. Likewise, increased plasma sEV concentrations were observed in other diseases associated with increased thrombosis risk, including thromboses associated with cancer [20] and cardiovascular diseases [21]. Additionally, enriched surface expression of CD133/1 and CD62P on sEVs from patients with APS could reflect the ongoing endothelial injury/activation and platelet activation, respectively, in these patients. Platelet activation was reported in APS, as measured by increased amounts of CD62P on these cells [10], and the aPL-driven thrombotic effects were shown to be mediated by CD62P [22]. CD133/1 has not been described in APS patients until now. 

Identification of surface markers on sEVs can reflect the cellular origin and molecular pathology in systemic autoimmune diseases [23,24]. Our study demonstrated the presence of platelet (CD41b and CD42a)-, CD8 lymphocyte-, endothelial cell (CD31)-, leukocyte (CD45)- and antigen presenting cell (HLA-ABC)-derived sEVs in plasma of patients with a history of thrombosis, as well as in HBD. These sEVs carried molecules involved in immune regulation (CD24, CD40, CD69, HLADRDPDQ), platelet/endothelial functions (CD62P, CD133/1), cell–cell adhesion (CD146, CD326) and extracellular matrix regulation (CD29, CD44, CD49e). Except for CD8, CD44, CD62P and CD133/1, these molecules were comparably expressed in all study groups. 

The T cell or NK cell origin of theCD8 positive sEVs remained unclear, since these sEVs were devoid of other detectable specific markers for these cells (CD3 and CD56, respectively), which was similarly observed by Koliha et al. [25]. There is a possibility that the assay was not sensitive enough to detect the low abundance surface proteins in the plasma sEV samples. Alternatively, some proteins may not be transported from the cells to the sEVs. Interestingly, we could not detect CD2-positive sEVs, usually present on T cells and NK cells, which is contrary to the findings of Koliha et al. [25]. One explanation could be that different ultracentrifugation protocols for sEV isolations were used in both studies.

Even though sEVs carrying the monocyte- and moDC surface markers were absent from our plasma samples, sEVs bearing HLA-ABC were detected in all three groups, indicating their origin from antigen-presenting cells.

Apart from sEVs originating from hematopoietic cells, we also found sEVs of endothelial cell origin, which did not significantly differ between the groups. However, significantly different expression of CD133/1 between aPL-neg IT patients and APS could indicate an altered endothelium in APS caused by aPL. 

CD44 has not yet been described on sEVs of APS patients. While sEVs bearing CD44 did not differ between APS patients and HBD, they were significantly decreased in aPL-neg IT patients, suggesting their reduced adhesion potential, circulation and homing of lymphocytes. 

## 5. Conclusions

A complex systemic network exists in the form of cell–cell communication via sEVs. In APS, this network could not only convey activation signals among cells, but could amplify the aPL-directed response from cell to cell, adding to the risk of thrombosis. Taken together, we observed higher numbers of sEVs in patients with APS and aPL-neg IT, suggesting an ongoing cell activation even in the absence of acute thrombotic events. Analysis of the surface protein profiles of sEVs has identified increased levels of CD8, CD62P, CD44 and CD133/1, among which CD8, CD44 and CD133/1 showed exclusive aPL-linked surface expression. sEVs from APS patients were specifically enriched in surface expression of CD62P, suggesting endothelial and platelet activation in APS. The release of CD62P-enriched sEVs might reflect augmented prothrombotic cellular activities in APS patients also in the absence of the acute thrombotic events. To our knowledge, this is the first study to characterize plasma sEVs in APS patients. This sets the basis for future sEV evaluation in larger patient cohorts and investigations into potential functional roles of sEVs in APS. 

## Figures and Tables

**Figure 1 cells-09-01211-f001:**
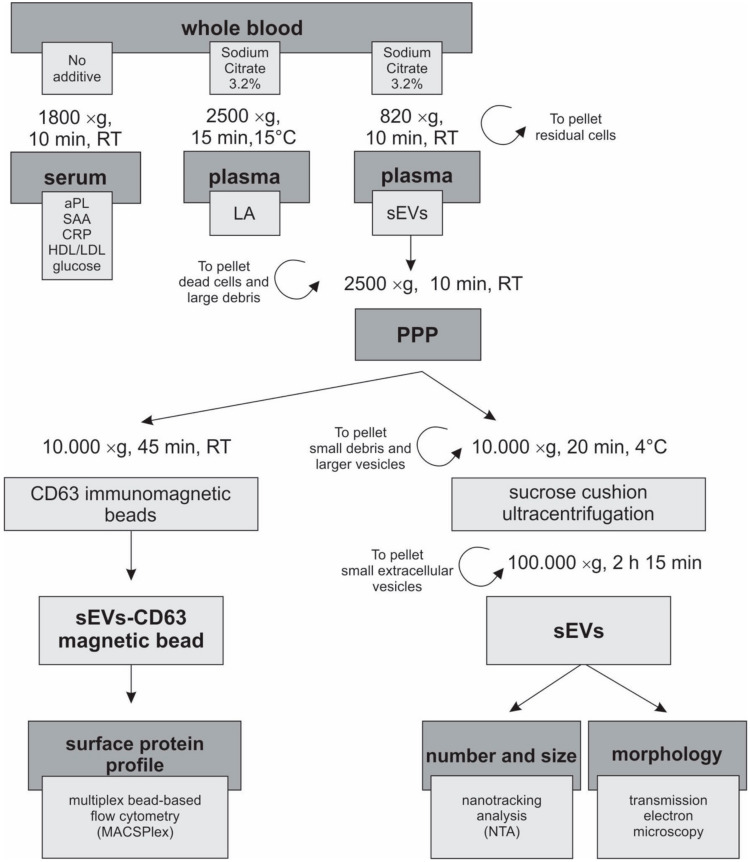
Chart of the sample preparation, procedure and analysis. Each participant had blood drawn into vacutainer tubes with either no additive or with 3.2% sodium citrate. Tubes were processed within one hour, carefully following the predefined procedure for isolation and characterization of sEVs. aPL, antiphospholipid antibody; CRP, C-reactive protein; HDL, high density lipoprotein; LA, lupus anticoagulant; LDL, low density lipoprotein; PPP, platelet-poor plasma; SAA, serum amyloid A; sEVs, small extracellular vesicles.

**Figure 2 cells-09-01211-f002:**
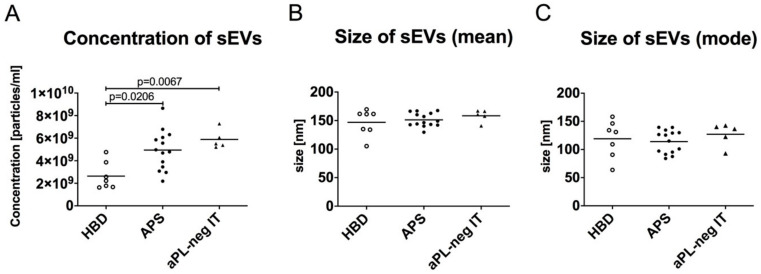
Concentration (**A**) and size (mean diameter (**B**) and mode diameter (**C**) of the sEVs isolated from the plasma of healthy blood donors (HBD), patients with antiphospholipid syndrome (APS) and aPL-neg patients with idiopathic thrombosis (aPL-neg IT). The sEVs were isolated from plasma using sucrose cushion ultracentrifugation and measured with nanoparticle tracking analysis (NTA). The nonparametric Kruskal–Wallis test with Dunn’s multiple comparison adjustment was used.

**Figure 3 cells-09-01211-f003:**
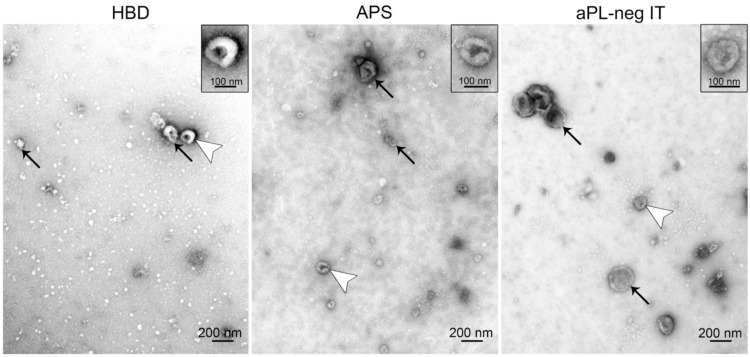
Representative transmission electron microscopy images of the sEVs isolated from the plasma of healthy blood donors (HBD), patients with antiphospholipid syndrome (APS) and aPL-neg patients with idiopathic thrombosis (aPL-neg IT). sEVs (arrows and arrowheads) exhibit the typical round shape morphology. Vesicles marked with white arrowheads are magnified in the upper right insets.

**Figure 4 cells-09-01211-f004:**
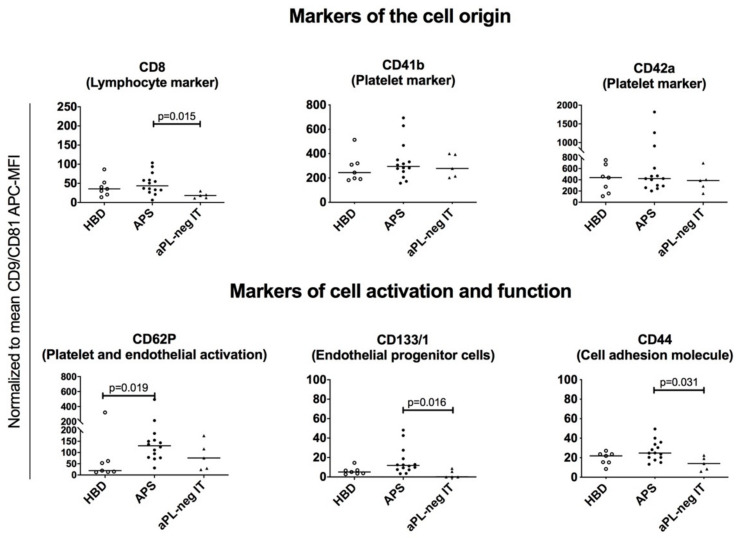
Normalized median fluorescence intensities (MFI) of the surface protein profiles of the plasma-derived sEVs from healthy blood donors (HBD), patients with antiphospholipid syndrome (APS) and aPL-neg patients with idiopathic thrombosis (aPL-neg IT). The nonparametric Kruskal–Wallis test with Dunn’s multiple comparison adjustment was used. Grouped surface protein profiles are shown indicating the cell of origin (upper panel) and cell activation status/functional properties of the sEVs (lower panel).

**Table 1 cells-09-01211-t001:** Patients’ demographics, clinical and laboratory features.

	HBD (*n* = 7)	APS (*n* = 14)	aPL-Neg IT (*n* = 5)	*p* Value
Mean age (range)	49 (28–67)	47 (28–74)	52 (38–73)	0.322
Sex (F:M)	6:1	9:5	3:2	0.534
Smoking	2/7 (29 %)	2/14 (14%)	1/5 (20%)	0.233
Fasting	1/7 (14%)	5/14 (36%)	2/5 (40%)	0.641
BMI (kg^2^)	24.1 ± 6.8	26.7 ± 4.0	24.3 ± 4.8	0.324
Systolic blood pressure (mmHg)	114.5 ± 9.4	137.8 ± 17.9	115 ± 15.2	0.005
Diastolic blood pressure (mmHg)	75.2 ± 5.2	84.5 ± 7.7	75.6 ± 11.0	0.029
Arterial thrombosis *n* (%)	0	6/14 (43%)	1/5 (20%)	0.363
Venous thrombosis *n* (%)	0	9/14 (64%)	4/5 (80%)	0.516
Microthrombosis *n* (%)	0	2/14 (14%)	0	0.372
Obstetric complications *n* (%)	0	3/14 (21%)	1/5 (20%)	ns
Diabetes *n* (%)	1/7 (14%)	2/14 (14%)	0	0.646
Anticoagulant therapy *n* (%)	0	12/14 (86%)	2/5 (40%)	0.005
Anti-aggregation therapy *n* (%)	0	4/14 (29%)	1/5 (20%)	0.471
Antimalarics *n* (%)	0	2/14 (14%)	0	0.266
Hormonal contraceptives *n* (%)	2/7 (29%)	4/14 (29%)	2/5 (40%)	0.624
aCL (G/M/A) *n* (%)	0	10/14 (71%)	0	0.001
IgG (<10 AU neg)	<5	20.9 ± 12.8	<5	0.001
IgM (<10 AU neg)	<5	10.4 ± 10.3	<5	0.157
IgA (<10 AU neg)	<5	4.4 ± 2.5	<5	0.145
anti-β2GPI (G/M/A) *n* (%)	0	11/14 (79%)	0	<0.001
IgG (<2 AU neg)	<2	10.9 ± 6.7	<2	0.002
IgM (<2 AU neg)	<2	2.21 ± 2.2	<2	0.081
IgA (<2 AU neg)	<2	1.9 ± 1.6	<2	0.333
aPS/PT (G/M/A) *n* (%)	0	11/14 (79%)	0	0.010
IgG (<5 AU neg)	<5	41.5 ± 44.6	<5	0.001
IgM (<5 AU neg)	<5	21.4 ± 29.1	<5	0.007
IgA (<5 AU neg)	<5	6.7 ± 5.6	<5	0.043
LA *n* (%)	/	10/14 (71%)	0	0.006

aCL, anti-cardiolipin antibodies; anti-β2GPI, anti-β2 glycoprotein I antibodies; aPS/PT, anti-phosphatidylserine/prothrombin antibodies; BMI, body mass index; IgG, immunoglobulin G; IgM, immunoglobulin M; IgA, immunoglobulin A; LA, lupus anticoagulant.

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
