# Peer review of "Characterization of Plasma-Derived Small Extracellular Vesicles Indicates Ongoing Endothelial and Platelet Activation in Patients with Thrombotic Antiphospholipid Syndrome"

_cells, 2020, doi:10.3390/cells9051211_

Round 1

Reviewer 1 Report

The manuscript of an original article entitled “Characterization of Plasma-Derived Small Extracellular Vesicles Indicates Ongoing Endothelial and Platelet Activation in Patients with Thrombotic Antiphospholipid Syndrome” by UlaŠtok and colleagues investigated the quantity, cellular origin and the cell surface expression of biologically active molecules in EVs from APS patients, aPL-negative patients with idiopatic thrombosis, and healthy controls. The study has very well characterized some basic information about circulating EVs in patients with APL. However, the study is still preliminary. It’s lack of association/correlation analysis, and mechanistic studies!

The presence of procoagulant condition and antibodies against phosphatidylserine/prothrombin are the common manifestations in APS. It has been well established in earlier studies (Liu ML, et al. Arterioscler Thromb Vasc Biol. 2007 Feb;27:430-5) in the EV research field, membrane surface phosphatidyl serine (PS, detection by annexin V) and tissue factor (TF) are crucial to the prothrombotic activity of EVs. In the current study, the authors want to address the role of sEVs in thrombotic APS. Therefore, detection of PS-positive, vs TF positive EVs may help the authors to address their points.

The Authors quantified the EV number, analyzed the cellular origin and the surface expression of biologically active molecules in small EVs (sEVs). This work may provide beneficial information in the mechanism of antiphospholipid syndrome and idiopathic thrombosis and may help the disease management in future. Association/correlation analysis between EVs and the prothrombotic or other disease conditions in the APL patients may also be helpful to explain their results from APL patients.

Some of the results need deeper discussion, i.e. the reason for increased blood pressure in APS patients than that of other control groups.

Reviewer 2 Report

Major and global comments:

The study reported in this article is very impressive by the oustanding technologies used, and the multiple parameters analyzed on the small Extracellular Vesicles (sEVs). I admire this huge work done. The authors did a considerable and team investigation, using very recently introduced technologies and they performed a deepful analysis on the characteristics of sEVs present in patients with APS, or with thrombotic diseases, as compared to a small group of healthy blood donnors.

However, this article contains too much information in a single report, too many parameters analyzed, and the major findings are not enough clearly presented: this decreases the message conveyed for readers. Furthermore, much information provided concerns basic biological/biochemical analyses in the tested population, and are not worth to be reported in detail. I strongly suggest to authors to review their report by focusing on the major information that they wish to share, and to structure their article accordingly. For example, the major finding concerns that patients with APS/aPL have many sEVs originated from endothelium. Results presented need to focus on proves of evidence on these EVs.

Presentation of some technical details can be summarized, and not detailed for hightlighting what is of essence for this original work, and reporting the findings of that study. Preparation of EVs, and the different centrifugation steps are a major information, as well as the specific CDs used for testing  the cell origin of sEVs. A table with the most characteristic targeted antigens/CDs for identificaztion of the cell origin should be advantageously presented in a table. Only figures showing the distribution of these sEVs are useful to be presented. But all the biochemical parameterds, patients' characteristics, and the excess of the many targeted CDs  could benefit from a much more simplified presentation. Authors should also indicate the main characteristics (relying mainly on vesicles's sizes) of EVs separated at each centrifugation/preparation step (820 g; 2,500 g; 10,000 g; 100,000 g).

The discussion section is too long and too disperse. It would better be condensed and focused on the major findings of that study.

Minor specific comments:

Line 27, abstract: it is not useful to remind APS here, as this sentence concerns only that group cited 2 lines before.

In introduction, line 49, please note that Antibodies are not directed to LA but their presence can induce LA activity.

What do the authors mean on lines 52-53 by: "presence of non criteria antibodies against phosphatidyl serine/prothrombine (aPS/PT)"?

In methods, the number of patients is low, and weakens the significance of that report. Especially, the number of healthy blood donors is very small (7). The control group is not matched correctly with APS or aPL neg-IT groups, at least in terms of gender distribution. This makes the comparison for some parameters of low significance on table 1.

On lines 129-130, please note that Stago LA is not proposed as an APTT reagent, but it is a reagent for detecting LA (inhibition of LA activity with hexagonal phase phospholipids).

In preparation of EVs (lines 132-140) it would be useful to indicate the size distribution of EVs in the various centrifugates/supernatants, and to show that the preparation method selects all small EVs (without loosing them in former steps).

On line 179, it should be "included" and not "includes".

Line 191, the authors probably mean: "...were corrected for background ...".

Table 1 is not necessary and reports too much information. It should be restricted to relevant analytes or cahracteristics differentiating groups. In addition, as there is a poor matching of pâthological groups with control group (more females in HBD), waist and hip circumference have no significance. In addition, the global reactivities of aPL assays is given, but this does not help to characterize patients' group. More information on aPL recativity of these APS patients should be useful.

Figure 3 is nice, but what is the information conveyed, which looks to be restricted to confirm the sEVs size? In legend to figure 3 it is indicated that  bars represent 100 nm, but on the picture it is actually 200 nm.

Results reported on figure 4 and pages 11 and 12 are too complex and too exhaustiuve, which impacts the majoir findings that the authors wish to point out.

Round 2

Reviewer 1 Report

No more comments

Author Response

Thank you for the comment. We have had the English language and style evaluated by a native speaker. Please find our corrections included in the tracked changes of the article.

Reviewer 2 Report

First of all, I would like to thank the authors for their extensive and well-documented answers to the various comments: these responses are all acceptable. The new version of the manuscript is clearly improved, and focuses better on the major findings of that stuy.

This report presents many results and it documents the type of EVs (especially the small ones) which can be found in APS compartively to Health Blood Donors, or to Idiopathic Thrombotic patients.

Minor remaining comments:

P2, line 51 (introduction): the authors have reminded the APS criteria, which is useful, but they should also include that the APS diagnosis is definite only when confirmed on 2 occasions at least 12 weeks apart.

P3, lines 83-91: it should be clearer to indicate the number of patients in each group by using figures rather then words (i.e. 19, 5 and 7).

P4, lines 133-135: Staclot LA is not an APTT reagent; it is only for LA detection/confirmation. It is not right to indicate that APTT was performed with Staclot LA. The authors probably mean that Staclot LA was used for detecting (or confirming) presence of LA (the hexagonal phase phospholipids present in Staclot LA reagent neutralise LA activity).

P 8, table 1: all the laboratory measurements reported (glucose, cholesterol, HDL, LDL, triglyceride, SAA, CRP, ESR, platelets) do not present significative differences between groups. It is not useful to excessively load this table with this information. It is enough to indicate in the results section that these variables were measured, but that there was no significant differences between groups.

P8, line 304, aPL-negative is repeated twice (suppress one).

Author Response

Please see the attachment. All lines and pages specified in the document refer to the manuscript without track changes. 
